# Which Factors Affect the Stress of Intraoperative Orthopedic Surgeons by Using Electroencephalography Signals and Heart Rate Variability?

**DOI:** 10.3390/s21124016

**Published:** 2021-06-10

**Authors:** Ji-Won Kwon, Soo-Bin Lee, Sahyun Sung, Yung Park, Joong-Won Ha, Gihun Kim, Kyung-Soo Suk, Hak-Sun Kim, Hwan-Mo Lee, Seong-Hwan Moon, Byung Ho Lee

**Affiliations:** 1Department of Orthopedic Surgery, Yonsei University College of Medicine, Seoul 03722, Korea; kwonjjanng@yuhs.ac (J.-W.K.); ghkim4201@yuhs.ac (G.K.); sks111@yuhs.ac (K.-S.S.); haksunkim@yuhs.ac (H.-S.K.); HWANLEE@yuhs.ac (H.-M.L.); SHMOON@yuhs.ac (S.-H.M.); 2Department of Orthopedic Surgery, Catholic-Kwandong University, Incheon 210-701, Korea; sumanzzz@ish.ac.kr; 3Department of Orthopedic Surgery, Ewha Womans University College of Medicine, Seoul 03760, Korea; sahyunsung@ewha.ac.kr; 4Department of Orthopedic Surgery, National Health Insurance Service Ilsan Hospital, Goyang 10444, Korea; yungpark@nhimc.or.kr (Y.P.); hjwspine@nhimc.or.kr (J.-W.H.)

**Keywords:** intraoperative stress, orthopedic surgery, surgeon, electroencephalography, heart rate variability, wearable device

## Abstract

Can we recognize intraoperative real-time stress of orthopedic surgeons and which factors affect the stress of intraoperative orthopedic surgeons with EEG and HRV? From June 2018 to November 2018, 265 consecutive records of intraoperative stress measures for orthopedic surgeons were compared. Intraoperative EEG waves and HRV, comprising beats per minute (BPM) and low frequency (LF)/high frequency (HF) ratio were gathered for stress-associated parameters. Differences in stress parameters according to the experience of surgeons, intraoperative blood loss, and operation time depending on whether or not a tourniquet were investigated. Stress-associated EEG signals including beta 3 waves were significantly higher compared to EEG at rest for novice surgeons as the procedure progressed. Among senior surgeons, the LF/HF ratio reflecting the physical demands of stress was higher than that of novice surgeons at all stages. In surgeries including tourniquets, operation time was positively correlated with stress parameters including beta 1, beta 2, beta 3 waves and BPM. In non-tourniquet orthopedic surgeries, intraoperative blood loss was positively correlated with beta 1, beta 2, and beta 3 waves. Among orthopedic surgeons, those with less experience demonstrated relatively higher levels of stress during surgery. Prolonged operation time or excessive intraoperative blood loss appear to be contributing factors that increase stress.

## 1. Introduction

The stress of workers in the medical field is undeniable and is an essential factor in healthcare quality management, given that despite the best of intentions and efforts patients’ health is not guaranteed. Surgery is a particularly stressful profession as it can affect the patient’s life and function given constant long working hours, highly intensive procedures, large learning curve required for conducting more than a certain degree [1,2,3]. During surgical procedures, the stress felt by surgeons may aid in the surgical process [2,4,5]. However, if the chronic stressful status of surgeon persists during surgery or there is a certain level of stress, it can be expected that catastrophic results such as technical problems, complexity of procedure, equipment failure, and patient complication may occur [1,3]. Perhaps the stress will increase more than the resting state of meditating peacefully, but we will not be able to know in real time the information on the extent and pattern of the increase. While various methods have been used to evaluate stress among surgeons, including questionnaires, heart rate, sympathovagal balance, heart rate variability (HRV), thermal activity, stress biomarkers in saliva, and smart patches [5,6,7,8,9,10], such methods do not accurately reflect the degree and type of stress in real-time and have limitations that are difficult to validate to the general public.

Therefore, we ultimately need to know to what extent intraoperative surgeon’s stress occurs in real time and how this stress can be related to clinical outcomes. As part of this, we have previously measured and reported the intraoperative stress analysis of orthopedic spine surgeons according to the role division (operator and assistant) and the stage of spine surgery (incision, instrumentation, decompression, and closure) [11]. However, orthopedic surgery is divided into subspecialties according to several anatomical regions, so there is a limitation that all orthopedic surgery cannot be defined as a general division. Beyond the scope of a previous study of the spine section, the need for increased generalization of the stress analysis of orthopedic surgeon has increased through extension to several orthopedic subspecialities(hip, knee, elbow, hand and ankle). And after dividing the cases belonging to several anatomic subspecialities into two types (tourniquet application, non-tourniquet application), we could investigate intraoperative surgeon’s stress according to the operating time and the intraoperative blood loss whether tourniquets were used.

This study aims to evaluate the real-time intraoperative stress analysis using wearable 2-channel EEG and heart rate variability detecting devices worn by orthopedic surgeons while performing surgery. We therefore asked:(1)Can we recognize intraoperative real-time stress in orthopedic surgeons?(2)Which factors affecting the stress of intraoperative orthopedic surgeons can be measured using EEG and HRV?

## 2. Materials and Methods

### 2.1. Subjects

This study was approved by the Institutional Review Board (IRB) of the corresponding authors’ hospital (Yonsei University Institutional Review Board and Ethics Committee: 4-2018-0363), with informed consent-for both study participation and publication of identifying information/images in and online open-access publication. All experiments were performed in accordance with The Code of Ethics of the World Medical Association (Declaration of Helsinki). From June 2018 to November 2018, 265 surgical stress records, including intraoperative EEG waves and HRV parameters, for eight orthopedic surgeons at three tertiary hospitals attached to a medical college were prospectively measured and analyzed. All eight surgeons were non-smokers and had no history of diseases such as hypertension, arrhythmia, ischemic heart disease, diabetes mellitus, and psychological disorders. None of the enrolled surgeons took beta-blockers or anti-arrhythmic medications that could affect their HRV parameters [6] and there is no overlap of consecutive records of orthopedic surgery at all of this study with the earlier study [11].

### 2.2. Materials

A two-channel EEG and HRV detecting wearable device (model: Amp GS5001; SOSO H&C, Kyungpook University, Daegu, Korea) was previously used for orthopedic spinal surgery and a psychiatric study [11,12]. This wearable device has attachment sites corresponding to Fp1 and Fp2 that reflect the brain’s frontal lobe cortical activity. Among the 21 classical EEG attaching nodes, it is possible to detect frontal cortex activity through Fp1 and Fp2 nodes, and it has the advantage of being comfortable to wear and not incurring cost. Two dry electrodes facing Fp1 and Fp2 enable wear while bound to one headband, and at the bottom of the mastoids on both sides. The EEG measurement of the frontal lobe may cause artifacts due to the wearer’s eyeball movement. As a solution to this, a neurologist specializing in EEG performed artifact rejection through visual inspection. The corrected EEG information for the artifact was re-analyzed using the MALTAB R2012b (MathWorks, Inc., Natick, MA, USA). Accordingly, using a fast Fourier transformation system, amplitudes for EEG data in the range of 1 to 50 Hz were calculated and aligned. The frequency power was calculated as the square of the amplitudes as follows: delta (0.5–3.5 Hz), theta (4–7 Hz), alpha (8–12 Hz), low beta (beta 1, 12–15 Hz), mid-range beta (beta 2, 15–20 Hz), high beta (beta 3, 20–30 Hz) and gamma waves (30–50 Hz) [13]. The calculated value for the EEG wave can be assumed as the ratio of the corresponding frequency powers to the whole frequency.

In terms of the HRV measurements, there is a small dry electrode corresponding to the ground and reference sensor so that it can perform simultaneous detection with heart beat variability [14]. There is a photoplethysmography (PPG) sensor inside the aforementioned dry electrode. When PPG light emitting diode is turned on, HRV measurements are started [15]. The sampling speed is 256 Hz, and online notch filtering is also applied at 60 Hz. The HRV was lineally recruited by moving window method to obtain HRVs according to the surgical progress [15]. A fast Fourier transform (FFT) was also used to calculate the power spectrum for extracting the features in the frequency domain.

### 2.3. Methods

Delta (0.5–3.5 Hz) waves are reported to be found during some continuous-attention tasks. Theta (4–7 Hz) waves reflect drowsiness and meditation, and increasing capabilities of executive functions. Alpha (8–12 Hz) waves reflect reduced anxiety, improving cognitive performance, and relaxation training for stress reduction. Low beta (beta 1, 12–15 Hz) waves are often associated with active, busy or anxious thinking and active concentration. Mid-range beta (beta 2, 15–20 Hz) waves are associated with increases in energy, anxiety, and performance. High beta (beta 3, 20–30 Hz) waves are associated with significant stress, anxiety, paranoia, high energy, and high arousal. And gamma waves (30–50 Hz) waves reflect carrying out a certain cognitive or motor function [13,16,17].

Factors affecting heart rate variability are largely divided into health conditions, heritance, physical condition, and stress. Therefore, increased activation levels in positive sympathetic dominance in the autonomic nervous system are considered a general counteracting reaction of our body to stress [7,18]. Subsequent intervals of five minutes were used for automatic calculation of the beats per minute (BPM) by an ear probe equipped with a PPG sensor. For time domain, square root of the mean R-R intervals (RMSSD) were used as parameters reflecting HRV. Low RMSSDs indicate stress [7,19,20]. Two main spectral densities were distinguished: the low frequency (LF) component (0.04–0.15 Hz) and the high frequency (HF) component (0.15–0.40 Hz). RMSSD, LF, and HF represent skewness of the autonomic nerve system, such as sympathetic and parasympathetic states. Stress is accompanied by increased LF, decreased HF, and increased LF/HF ratios [6,10,18,21].

The device was worn preoperatively and EEG signals and HRV recordings began prior to hand-washing for surgery and ended at the conclusion of surgery (Figure 1). Data were recorded in the progress note of the surgeons in each stage, including 1st, 2nd, and 3rd stages to confirm exact time flow during the surgery. The 1st stage began with the aseptic method of wiping hands and elbows, donning of surgical gown, and draping the surgical window. The 2nd stage is generally defined as the climax including the main course of orthopedic surgical procedure. Finally, the 3rd stage entails steps to complete the surgical treatment. The EEG and HRV in the resting state were also measured with the eyes closed for 3 min, and were used as a baseline for comparison.

In orthopedic surgery, to reduce bleeding and to facilitate surgical window exposure, a tourniquet that blocks the circulation of blood vessels in the proximal to the surgical site during working time may be applied [22,23,24]. However, in spine or hip surgeries, tourniquets cannot be applied due to the anatomic regions; such procedures tend to result in greater blood loss. Thus, procedures were divided into two groups depending upon tourniquet use. Surgeons’ EEG and HRV were evaluated according to the amount of bleeding and operation time of each group after surgery. Then, intraoperative data were compared with that of the resting status. In addition, eight orthopedic surgeons were divided into three groups according to orthopedic surgery experience: <5 years, 5–15 years, and >15 years [4,25,26]. EEG and HRV values were evaluated according to clinical experience and the ratio of EEG and HRV during surgery was compared to resting status.

### 2.4. Statistical Analyses

One-way analysis of variance (ANOVA) was used to evaluate significant differences in measurements according to the experience of surgeon. After dividing into subgroups according to surgical experience, in order to reduce replication of positive result, each surgical case was scaled by the ratio of stress-associated EEG and HRV parameters to baseline EEG and HRV parameters measured before each surgical case. The parameters corresponding to each surgical stage classified chronologically using group as a factor were compared and analyzed by One-way analysis of variance (one-way ANOVA). Bonferroni post-hoc analyses were used to confirm significant between-group differences. Bivariate Pearson correlation analysis was used to describe correlations in intraoperative stress with possible influencing parameters, such as intraoperative bleeding, and operation time. All statistical analyses were performed using the SPSS 22.0 statistical package (SPSS, International Business Machines Corp., Armonk, NY, USA). *p*-values < 0.05 were considered statistically significant.

## 3. Results

### 3.1. Demographics

Intraoperative stress using wearable EEG and HRV detecting device was measured in 265 cases, and eight orthopedic surgeons were enrolled. Demographic data including each surgeon’s experience, subspecialty, age, sex, accumulated surgical experience, and any underlying health conditions are shown in Table 1.

There were a total of 102 spine and hip orthopedic surgery cases, which did not employ tourniquets, and the mean operation time and intraoperative blood loss were 175.1 ± 75.5 min and 436.7 ± 542.6 cc, respectively. There was a total of 163 surgical cases of subspecialty surgeries using tourniquets, including foot and ankle, hand and wrist, elbow, and knee. Mean operation time and intraoperative blood loss were 121.6 ± 65.8 min and 55.2 ± 46.3 cc. The 265 cases were divided into three groups according to surgeons’ experience: Group A < 5 years, Group B 5–15 years, and Group C > 15 years. Group A included 100 surgeries, performed by three novice surgeons. Subspecialties included was spine, hand and wrist, and foot and ankle. Group B included 100 surgeries, performed by three associate professor surgeons. Subspecialties included spine, hip, and knee. Group C included 65 surgeries performed by two senior surgeons. Subspecialties included knee and elbow. The operation time and intraoperative blood loss of Group A, B, and C were 166.1 ± 90.3 min and 289.1 ± 526.4 cc, 143.3 ± 64.6 min and 209.7 ± 308.7 cc, and 103.7 ± 35.7 and 54.9 ± 44.0 cc, respectively.

### 3.2. Stress Analysis According to the Experience of Surgeons

For group A surgeons, all parameters varied significantly between stages, except for beta 2, alpha waves, and BPM. (*p* < 0.001, ANOVA). Beta 3 levels and gamma waves remained high in all stages compared to groups B and C. Bonferroni analysis indicated that beta 3 and gamma waves were highest during the 2nd stage, but decreased as the surgery progressed (*p* = 0.014, *p* = 0.042, respectively). In addition, in group A, the LF/HF ratio remained high in the 1st stage. As the procedure progressed, the LF/HF of group A decreased to a ratio close to one. Among group B, all parameters changed significantly during surgery, except for RMSSD. (*p* < 0.001, ANOVA). The EEGs of group B changed significantly during surgery, except for delta wave (*p* < 0.001, ANOVA). Bonferroni analysis indicated that beta 1 waves were highest during the 1st stage. Gamma waves peaked at a ratio greater than one in the 2nd stage. The beta 3 waves displayed a pattern similar to that of the gamma wave, but not significantly higher than the resting state. In the 1st and 2nd stages, the LF/HF ratio maintained a value greater than two. In the 3rd stage, the ratio value decreased to ~1.5. EEGs of group C, unlike groups A and B, displayed a somewhat flat pattern, with no significant changes across stages. LF/HF ratio increased from 1st to 3rd stage, with the higher levels of LF/HF ratio in the 3rd stage (*p* = 0.002) (Table 1, Figure 2).

### 3.3. Subgroup Analysis Whether to Use Tourniquet or Not

During the non-tourniquet orthopedic surgeries, operation time was not significantly correlated with the corresponding EEG and HRV parameters of 1st, 2nd, and 3rd stage. Accumulated intraoperative blood loss was positively correlated with beta 1, 2, and 3 waves. Also, intraoperative blood loss was negatively correlated with gamma waves in 1st, 2nd, and 3rd stage. HRV parameters including RMSSD were negatively correlated with accumulated intraoperative blood loss in the 1st and 3rd stage. (*p* < 0.05) (Table 2). In surgeries with tourniquets, operation time was positively correlated with beta 1, 2, and 3 waves. Unlike non-tourniquet surgeries, there was no significant correlation with intraoperative blood loss. HRV parameters including BPM and LF/HF were positively correlated with accumulated intraoperative blood loss and operation time in the 1st, 2nd, and 3rd stage. (*p* < 0.05) (Table 3 and Table 4).

## 4. Discussion

The first main finding was that the beta waves were higher among novice surgeons than experienced surgeons, indicating that the psychological stress decreases with surgical experience. Additionally, considering maintaining a relatively high beta 3 waves value in the 1st and 2nd stage, it can be predicted that the initial stress of the novice surgeon may be high when anticipating the course of surgical treatment and planning the process. Beta waves for the more experienced surgeons tended to increase in the 2nd stage where the main course of surgery is concentrated. However, the beta waves were significantly lower at this time than at rest, which may indicate that this rather repetitive and familiar surgical technique is not a mental stress for expert surgeons [27]. As the 3rd stage progressed, the heart rate variability parameters were higher among experienced surgeons, and it can be predicted that the physical stress is felt at a higher age [7,20]. It is conceivable that the heart rate variability depicting the anti-cholinergic autonomic nervous system can be a more significant stress marker for experienced surgeons [18,20]. The more experienced surgeons tend to be older. Aging is associated with baseline changes in HRV. Thus, there might be age-related changes that could affect the baseline and response during surgery.

In the surgical group who used tourniquets, regardless of intraoperative blood loss, the longer the operation time, the higher the beta waves were in EEGs, suggesting that the surgeons’ stress related to concentration increased when the operation time was extended beyond the average over time. Operations lasting longer than anticipated may affect the surgeon’s anxiety and increase physical stress, suggesting that this affects the aforementioned HRV parameters [1,5,28]. In the surgeries without tourniquets, the greater the intraoperative blood loss, the greater the proportion of beta waves in the total EEG frequencies of surgeons. In orthopedic surgery, hip arthroplasty, shoulder arthroplasty, and spine surgery are typically included in the non-tourniquet group [29]. This implies that the increase in intraoperative blood loss can be a surrogate marker representing the difficulty of surgery, such that stress can increase when surgeons encounter increasing surgical difficulty [1,3,5]. These changes occurred generally in the 3rd stage, and since the degree of intraoperative blood loss can be estimated mainly in the last stage, this change can be anticipated. HRV parameters are also mainly concentrated in the 3rd stage, and tended to increase relative to baselines [2].

In general, it is easy to deduce that the surgeon’s stress increases when the surgeon’s experience is insufficient or when the operation time and intraoperative blood loss increase. However, specifying these predictions with actual numerical values and parameters that have been validated has another important meaning. Numerous studies have been conducted on several factors related to the surgical outcome [30,31,32]. These are the demographic data of the patients before surgery, radiologic parameters of the patients undergoing surgery, and factors related to surgery such as operation time and intraoperative blood loss. The relationship between these factors and the clinical or radiologic outcome of patients after surgery has been an issue and concern that has been studied in all academic fields. However, among these factors, studies on the relationship between surgeon’s stress during surgery are scarce. Indeed, many studies have not been conducted and focused on the stress of medical workers responsible for the health of patients. If large-scale data on surgeons’ stress accumulates, and the correlation between postoperative outcome and surgeon’s stress during surgery can be identified, this study could be a motive that has clinical significance.

The present study has limitations. We did not evaluate surgeons’ handedness and more extensive symptoms, such as severity of inattention and burn-out status using questionnaires. Additionally, resting status EEG was not assessed during sleep. Further, we measured several parameters that represent surgeons’ stress over time. If an event that causes surgeons’ stress to increase is concentrated in a specific time period, there may be a bias in measurement by region divided by time period. In this study, all surgeons were male, therefore, generalization to female surgeons should be made with caution. Also, this study was not carried out by recruiting a large number of surgeons. In order to reduce the replication of positive values, the EEG and HRV parameters measured in each surgical case were measured separately from the baseline values measured before surgery and proceeded in the form of a ratio. Therefore, it is necessary to be careful not to over-interpret the results. Finally, despite being a preliminary study, we could not even compare the surgeon’s stress-associated parameters with many variables other than intraoperative blood loss and operation time. Hence, future researchers should be aware of these limitations and consider larger sample sizes and recruiting control groups matched for sex, age range, anxiety symptoms, and length of experience.

## 5. Conclusions

Among orthopedic surgeons, those with little experience demonstrated relatively higher levels of stress-related EEG waves compared to resting status during surgery. HRV parameters indicating physically demanding stress remained high as surgical treatment progressed over time in senior orthopedic surgeons. Prolonged operation time or excessive intraoperative blood loss could be assumed to be contributing factors that increase surgeons’ stress-associated parameters depending upon whether tourniquets are used.

## Figures and Tables

**Figure 1 sensors-21-04016-f001:**
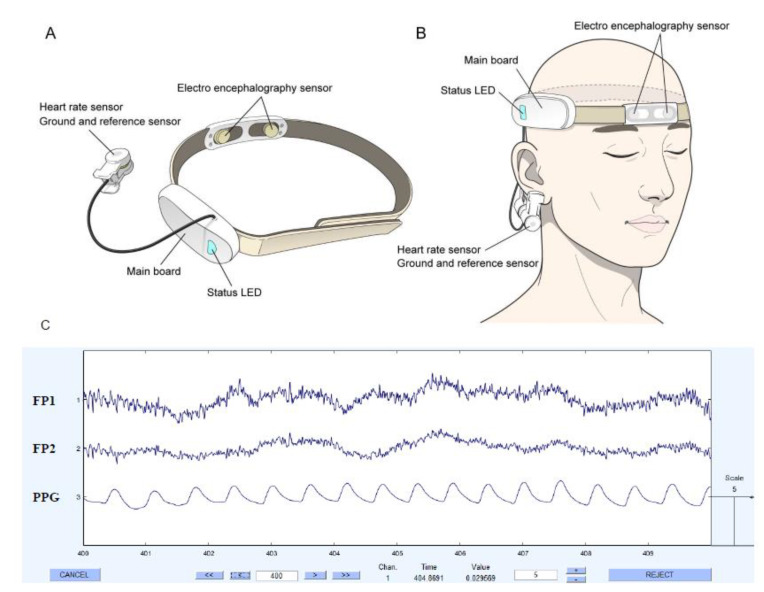
(**A**) Profile of the wearable device. (**B**) The device is worn on the head. (**C**) Captured the raw signals displayed on MALTAB software. EEG; Electroencephalography, PPG; Photoplethysmography.

**Figure 2 sensors-21-04016-f002:**
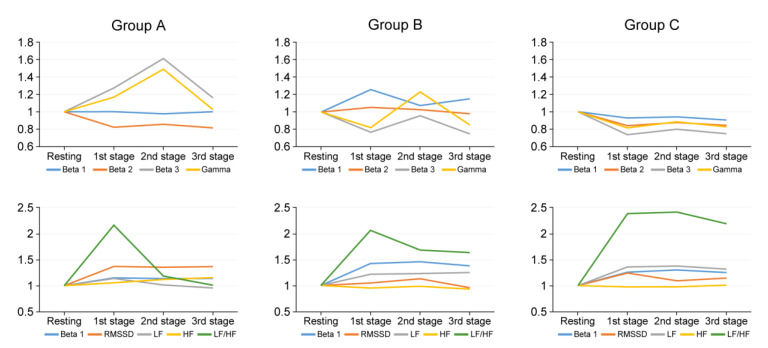
Pattern of stress related EEG waves and HRV parameters according to the experience of surgeon and three stages over time during surgical procedure. According to experience of surgeon; Group A, less than 5 years; Group B, from 5 years to less than 15 years; Group C, more than 15 years; Beta 1, low beta waves, 12–15 Hz; Beta 2, mid-range beta waves, 15–20 Hz; Beta 3, high beta waves, 20–30 Hz; Gamma, 30–50 Hz. BPM: beats per minute, RMSSD: square root of the mean R-R interval; LF: low frequency, HF: high frequency.

**Table 1 sensors-21-04016-t001:** Demographic Characteristics of the Enrolled Surgeons. Baseline of the EEG is presented as ratio per whole frequency EEG. Beta 1: low beta waves, 12–15 Hz; Beta 2: mid-range beta waves, 15–20 Hz; Beta 3: high beta waves, 20–30 Hz; Gamma; 30–50 Hz; BPM: beats per minute, RMSSD: square root of the mean R-R interval; LF: low frequency, HF: high frequency. Baseline of the EEG was presented as ratio per whole EEG frequency. Values were presented as mean ± standard deviation.

Surgeon	1	2	3	4	5	6	7	8
Age (years)	43	35	35	36	43	56	50	38
Experience in orthopedic subspecialty (years)	10	5	2	3	9	23	16	4
Orthopedic subspecialty	Spine	Hip	Spine	Hand and wrist	Knee	Knee	Elbow	Foot and ankle
Enrolled cases (cases)	35	35	32	33	30	32	33	35
Whether to use tourniquet	No	No	No	Yes	Yes	Yes	Yes	Yes
Baseline of the EEG (resting status)								
Beta 1 wave	0.05 ± 0.004	0.05 ± 0.004	0.03 ± 0.005	0.05 ± 0.001	0.06 ± 0.006	0.04 ± 0.004	0.03 ± 0.045	0.04 ± 0.005
Beta 2 wave	0.07 ± 0.009	0.08 ± 0.006	0.07 ± 0.008	0.09 ± 0.003	0.10 ± 0.002	0.08 ± 0.011	0.06 ± 0.010	0.09 ± 0.010
Beta 3 wave	0.13 ± 0.017	0.16 ± 0.009	0.15 ± 0.015	0.05 ± 0.011	0.15 ± 0.009	0.17 ± 0.022	0.14 ± 0.178	0.14 ± 0.020
Gamma wave	0.15 ± 0.021	0.19 ± 0.135	0.21 ± 0.026	0.10 ± 0.017	0.15 ± 0.004	0.18 ± 0.025	0.16 ± 0.217	0.13 ± 0.028
Baseline of the HRVs (resting status)								
BPM	58.0 ± 4.60	76.9 ± 8.48	76.4 ± 8.30	83.0 ± 4.42	59.4 ± 4.94	86.0 ± 4.80	61.4 ± 4.52	83.7 ± 6.35
RMSSD	41.9 ± 3.74	37.9 ± 5.89	39.5 ± 5.87	26.1 ± 4.63	27.3 ± 0.70	34.7 ± 7.46	23.5 ± 2.07	21.2 ± 3.91
LF/HF	1.2 ± 0.55	0.4 ± 0.84	0.7 ± 0.84	0.4 ± 0.32	0.6 ± 0.31	0.5 ± 0.46	1.3 ± 0.81	1.4 ± 0.34

**Table 2 sensors-21-04016-t002:** Characteristics of groups classified into experience in orthopedic subspeciality. Group A, less than 5 years; Group B, from 5 years to less than 15 years; Group C, more than 15 years.

	Group A	Group B	Group C
Surgeons	No 3, 4, 8	No 1, 2, 5	No 6, 7
Mean Age (years)	36.3	40.3	53.0
Enrolled cases (cases)	100	100	65
Orthopedic subspecialty	spine, hand and wrist, and foot and ankle	spine, hip, and knee	knee and elbow
Operation time (minutes)	166.1 ± 90.3	143.3 ± 64.6	103.7 ± 35.7
Intraoperative blood loss (cc)	289.1 ± 526.4	209.7 ± 308.7	54.9 ± 44.0

**Table 3 sensors-21-04016-t003:** Correlation Analysis Between the Stress-Related Parameters of the Surgeon in Non-Tourniquet Applied Orthopedic Surgery. * *p* < 0.05, ** *p* < 0.01 Beta 1: low beta waves, 12–15 Hz; Beta 2: mid-range beta waves, 15–20 Hz; Beta 3: high beta waves, 20–30 Hz; Gamma; 30–50 Hz; BPM: beats per minute, RMSSD: square root of the mean R-R interval; LF: low frequency, HF: high frequency.

	Beta 1	Beta 2	Beta 3	Gamma	BPM	RMSSD	LF/HF
1st stage							
Operation time	−0.004	−0.086	−0.082	−0.178	−0.097	0.233 *	−0.053
Accumulated intraoperative blood loss	0.325 **	0.251 *	0.385 **	−0.452 **	0.148	−0.227 *	−0.042
2nd stage							
Operation time	−0.053	−0.167	−0.197	−0.233 *	−0.101	0.225 *	−0.041
Accumulated intraoperative blood loss	0.357 **	0.288 **	0.373 **	−0.393 **	−0.142	−0.148	−0.147
3rd stage							
Operation time	−0.043	−0.071	−0.013	−0.032	−0.089	0.158	−0.001
Accumulated intraoperative blood loss	0.411 **	0.366 **	0361 **	−0.431 **	−0.061	−0.201 *	−0.308 **

**Table 4 sensors-21-04016-t004:** Correlation Analysis Between the Stress-Related Parameters of the Surgeon in Tourniquet Applied Orthopedic Surgery. * *p* < 0.05, ** *p* < 0.01 Beta 1: low beta waves, 12–15 Hz; Beta 2: mid-range beta waves, 15–20 Hz; Beta 3: high beta waves, 20–30 Hz; Gamma: 30–50 Hz; BPM: beats per minute, RMSSD: square root of the mean R-R interval; LF: low frequency, HF: high frequency.

	Beta 1	Beta 2	Beta 3	Gamma	BPM	RMSSD	LF/HF
1st stage							
Operation time	0.308 **	0.501 **	0.303 **	0.042	0.303 **	0.035	−0.046
Accumulated intraoperative blood loss	−0.032	0.058	0.047	0.001	0.109	−0.385 **	0.348 **
2nd stage							
Operation time	0.289 **	0.372 **	0.094 *	−0.305 **	0.259 **	0.048	−0.008
Accumulated intraoperative blood loss	−0.049	0.202 **	0.081	0.072	0.339 **	−0.425 **	0.432 **
3rd stage							
Operation time	0.115	0.363 **	0.171 *	−0.013	0.205 **	0.002	−0.142
Accumulated intraoperative blood loss	−0.254 **	−0.024	0.066	0.068	0.202 **	−0.384 **	0.333 **

## Data Availability

The data presented in this study are available on request from the corresponding author.

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
