# Peer review of "Which Factors Affect the Stress of Intraoperative Orthopedic Surgeons by Using Electroencephalography Signals and Heart Rate Variability?"

_sensors, 2021, doi:10.3390/s21124016_

Round 1
Reviewer 1 Report
The article is too brief and not detailed enough. This aspect should be improved.
However, I prefer to start with "plagiarism" or rather, ethical concerns. I have reservations that it is enough different from [1], so the authors should state more clearly what is new with respect to it.
Returning to the reviewed article, the introduction is too short, please expand it. Lack of information on other similar research studies, for comparison.
In Section 2, I suggest changing the subtitles "A wearable device detecting intraoperative EEG and HRV of orthopedic surgeons" and "Evaluation of intraoperative stress via EEG and HRVs" to "Materials" and "Methods" respectively; removing the following subtitles from this section. Duplicate explanations should also be eliminated.
I also have problems with the ECG: It only shows where the reference electrode is, but what about the other electrodes? Which lead is used? Some of the captured raw signals should be displayed (both, EEG and ECG).
Another problem is that 9 volunteers divided into 3 subgroups seems too many segmentations, even for a pilot study, which is not a large enough sample to draw conclusions. Please explain these limitations.
The results section lacks informative tables on mean values, standard deviations, etc. for each group and variable used.
The discussion should be modified to include all new elements, and the conclusions should be improved.
[1] https://doi.org/10.1016/j.spinee.2020.02.006
Author Response
Reviewer 1
The article is too brief and not detailed enough. This aspect should be improved.
Before addressing your comment, I would like to express my appreciation for your efforts in reviewing our manuscript. Through the process of revising the thesis according to your point of view, the completeness has improved a lot. From the author's point of view, one kind of preliminary study could have the significance of clarifying the report on new facts to readers. In that sense, we faithfully acknowledge and respect the importance of the reviewing process. Many thanks for your efforts.
However, I prefer to start with "plagiarism" or rather, ethical concerns. I have reservations that it is enough different from [1], so the authors should state more clearly what is new with respect to it.
[Therefore, we ultimately need to know to what extent intraoperative surgeon's stress is in real time and how this stress can be related to clinical outcomes. As part of this, we have previously measured and reported the intraoperative stress analysis of orthopedic spine surgeon according to the role division (operator and assistant) and the stage of spine surgery (incision, instrumentation, decompression, and close) [11]. However, orthopedic surgery is divided into subspecialties according to several anatomical regions, so there is a limitation that all orthopedic surgery cannot be defined as a general division. Beyond the scope of a previous study of the spine section, the need for increased generalization of the stress analysis of orthopedic surgeon has increased through extension to several orthopedic subspecialities(hip, knee, elbow, hand and ankle). And after dividing the cases belonging to several anatomic subspecialities into two types (Torniquet apply, non-tourniquet apply), we could investigate intraoperative surgeon's stress according to the operating time and the intraoperative blood loss whether torniquets were used.]
In your opinion, this study is a study conducted on a completely different database from previous studies, and has new information. As you pointed out, I explained it in more detail in the introduction. I want to combat the unethical practice of redundant publications. Therefore, I wanted to increase the credibility of the paper by addressing ethical issues that may be of concern in advance. I express my deep respect for your understanding of my intentions.
Returning to the reviewed article, the introduction is too short, please expand it. Lack of information on other similar research studies, for comparison.
[The stress of workers in the medical field is undeniable and is an essential factor in quality management, given that despite the best of intentions and efforts patients’ health is not guaranteed. Surgery is a particularly stressful profession as it can affect the patient's life and function given constant long working hours, highly intensive procedures, large learning curve required for conducting more than a certain degree [1-3]. During surgical procedures, the stress felt by surgeons may aid in the surgical process [2,4,5]. However, if the chronic stressful status of surgeon persists during surgery or there is a certain level of stress, it can be expected that catastrophic results such as technical problems, complexity of procedure, equipment failure, and patient complication may occur [1,3]. Perhaps the stress will increase more than the resting state of meditating peacefully, but we will not be able to know in real time the information on the extent and pattern of the increase. While various methods have been used to evaluate stress among surgeons, including questionnaires, heart rate, sympathovagal balance, heart rate variability (HRV), thermal activity, stress biomarkers in saliva, and smart patches [5-10], such methods do not accurately reflect the degree and type of stress in real-time and have limitations that are difficult to validate to the general public….]
According to your opinion, the content of the introduction has been reinforced and attached.
In Section 2, I suggest changing the subtitles "A wearable device detecting intraoperative EEG and HRV of orthopedic surgeons" and "Evaluation of intraoperative stress via EEG and HRVs" to "Materials" and "Methods" respectively; removing the following subtitles from this section. Duplicate explanations should also be eliminated.
I changed the subtitles to "Materials" and "Methods" according to your opinion. And, for the duplicate explanation, the repeated contents were changed again. Thank you.
I also have problems with the ECG: It only shows where the reference electrode is, but what about the other electrodes? Which lead is used? Some of the captured raw signals should be displayed (both, EEG and ECG).
Figure 1. A. Profile of the wearable device. B. The device is worn on the head. C. Captured the raw signals displayed on MALTAB software. EEG ; Electroencephalography, PPG ; Photoplethysmography
[In terms of the HRV measurements, there is a small dry electrode corresponding to the ground and reference sensor so that it can perform simultaneous detection with heart beat variability. There is a PPG(Photoplethysmography) sensor inside the aforementioned dry electrode. When PPG light emitting diode is turned on, HRV measurements are started [14]. The sampling speed is 256 Hz, and online notch filtering is also applied at 60 Hz. The HRV was lineally recruited by moving window method to obtain HRVs according to the surgical progress [14]. And fast Fourier transform (FFT) was also used to calculate the power spectrum for extracting the features in the frequency domain.]
Thanks for your comment. The captured raw signal is attached in Figure 1 to facilitate the reader's understanding. In addition, heart rate variability was corrected as data through light emitting diodes through PPG (Photoplethysmography). This is detailed in the Material section.
Another problem is that 9 volunteers divided into 3 subgroups seems too many segmentations, even for a pilot study, which is not a large enough sample to draw conclusions. Please explain these limitations.
- Broeders, J.A.; Draaisma, W.A.; Impact of surgeon experience on 5-year outcome of laparoscopic Nissen fundoplication. Archives of surgery (Chicago, Ill. : 1960) 2011, 146, 340-346.
- Forbes, T.L. A cumulative analysis of an individual surgeon's early experience with elective open abdominal aortic aneurysm repair. American journal of surgery 2005, 189, 469-473.
- Cahill, P.J.; Youth and Experience: The Effect of Surgeon Experience on Outcomes in Cerebral Palsy Scoliosis Surgery. Spine deformity 2018, 6, 54-59.
The age of the volunteers varies, and as you pointed out, the number of enrolled orthopedic surgeon is not large. Therefore, referring to the references mentioned above, I tried to demonstrate the results of stress-associated EEG & physically demanded autonomous stress-associated HRV by dividing surgical experience into three stages. In your opinion, there are not many surgeons. However, the number of surgeries performed by individual surgeons can be seen as a surgical case that is not small. And, for a more accurate comparison, baseline EEG and HRV were measured, and the results of the stress-associated parameters according to the operation were compared to the baseline.
The results section lacks informative tables on mean values, standard deviations, etc. for each group and variable used.
As you pointed out, I added baseline individual EEG and HRV data to the table for informative data. In addition, a table representing each group (A, B, and C) was added to make it easier for readers to understand.
The discussion should be modified to include all new elements, and the conclusions should be improved.
As you said, the discussion section has been reinforced and the conclusion has been revised.

Reviewer 2 Report
This study examines factors that may explain increased stress in orthopedic surgeons during surgical procedures. These data are unique; however, there are concerns particularly with the statistical analyses outlined below,
- The rationale for the study needs better explanation in the introduction. The connection between stress and poor surgical outcomes seems weak. Are there references to include that support this assumption? One could argue that increased stress may improve focus and performance, thereby improving clinical outcomes.
- In the statistical analyses section, a brief description of how and why the data were scaled should be included. Including details on the model of ANOVA used and factors would be helpful.
- The use of ANOVA is questionable and requires further explanation or justification. The statistical analyses appear to use several cases (n = 256) for a small number of surgeons (n = 8). Surgeons (and cases) are further divided into 3 groups based on experience. Because the same surgeon is performing multiple cases, each case or observation will not be independent, thereby violating a core assumption of ANOVA. Thus, it appears that the degrees of freedom (which are based on cases?) are inflated due to replication, thereby increasing Type I error.
- The conclusions advanced in lines 323-328 seem to focus on differences between surgeons. This does not seem appropriate considering that only 8 surgeons (divided into 3 groups) were enrolled in the study.
- The more experienced surgeons tend to be older. Aging is associated with baseline changes in HRV. Thus, there may be age-related changed that may affect baseline and hence responses during surgery. This may be worth addressing in the discussion.
- It seems unnecessary to include the last 3 rows of Table 1 considering that surgeons do not differ on these variables and is already mentioned in the text of the results. Rather, it may be useful to provide baseline values of heart rate and some of the stress measures.
- Is the use of the words “real-time (assessment of stress)” accurate considering that a 5-min recording is used for HRV? Also, since the measure is not more instantaneous how do HRV values have utility for a surgeon?
- What criteria was used in selecting a representative ECG segment? How were ectopic beats handled? Describing the criteria that were used is important to the validity of the research findings.
- The 2nd and 3rd sentence of the abstract should be edited for sentence structure.
- The first paragraph of the discussion does not seem to add much to this section and could be deleted. The 3rd paragraph of the discussion that begins with “The first main finding…” may be more effective if it moved up to the 1st paragraph of the discussion.
- In line 133 low RMSSD is said to reflect increased stress whereas in line 139 stress is accompanied by increased RMSSD.
- A reference that supports the validity of using earlobe pulse for heart rate variability in short-term recordings would be important to include in the methods. Are their limitations to address with this technique such as influence of respiratory rhythm?
- References ‘1’ and ‘2’ needs revision in the references list.
- Were there unanticipated events that occurred during any of the surgeries? What was the associated change in HRV and EEG patterns under these conditions? This may provide additional strength to the role of surgeon experience and ‘staying cool under pressure’.
Author Response
Reviewer 2
This study examines factors that may explain increased stress in orthopedic surgeons during surgical procedures. These data are unique; however, there are concerns particularly with the statistical analyses outlined below,
Before replying to your comment, thank you for your enormous efforts in the reviewing process.
- The rationale for the study needs better explanation in the introduction. The connection between stress and poor surgical outcomes seems weak. Are there references to include that support this assumption? One could argue that increased stress may improve focus and performance, thereby improving clinical outcomes.As you mentioned, stress can also have a positive effect on performance, such as surgery. So far, there have been no reports to elucidate the relationship between worker (surgeon) stress and clinical outcomes of patients. Moreover, there are few reports on the stress during operation of workers (surgeons). We started this study to explore unknown areas by measuring the surgeon's EEG and HRV in real time. In addition, according to your comment, the contents of the meaning of this study and the clinical usefulness to be obtained in the future have also been added to the introduction section. Thank you.
- [The stress of workers in the medical field is undeniable and is an essential factor in quality management, given that despite the best of intentions and efforts patients’ health is not guaranteed. Surgery is a particularly stressful profession as it can affect the patient's life and function given constant long working hours, highly intensive procedures, large learning curve required for conducting more than a certain degree [1-3]. During surgical procedures, the stress felt by surgeons may aid in the surgical process [2,4,5]. However, if the chronic stressful status of surgeon persists during surgery or there is a certain level of stress, it can be expected that catastrophic results such as technical problems, complexity of procedure, equipment failure, and patient complication may occur [1,3]. Perhaps the stress will increase more than the resting state of meditating peacefully, but we will not be able to know in real time the information on the extent and pattern of the increase. While various methods have been used to evaluate stress among surgeons, including questionnaires, heart rate, sympathovagal balance, heart rate variability (HRV), thermal activity, stress biomarkers in saliva, and smart patches [5-10], such methods do not accurately reflect the degree and type of stress in real-time and have limitations that are difficult to validate to the general public…]
- In the statistical analyses section, a brief description of how and why the data were scaled should be included. Including details on the model of ANOVA used and factors would be helpful. After dividing into subgroups according to surgical experience, each surgical case was scaled by the ratio of stress-associated EEG and HRV parameters to baseline EEG and HRV parameters measured before each surgical case. The parameters corresponding to each surgical stage classified chronologically using group as a factor were compared and analyzed by One-way analysis of variance (one-way ANOVA).
- Thanks for your comment. As you said, we added a brief description to the statistical analyzes section.
- The use of ANOVA is questionable and requires further explanation or justification. The statistical analyses appear to use several cases (n = 256) for a small number of surgeons (n = 8). Surgeons (and cases) are further divided into 3 groups based on experience. Because the same surgeon is performing multiple cases, each case or observation will not be independent, thereby violating a core assumption of ANOVA. Thus, it appears that the degrees of freedom (which are based on cases?) are inflated due to replication, thereby increasing Type I error.
Thanks for your point. As you said, if the same surgeon performs multiple cases at once, I think it can have a reciprocal effect on successive cases. This study is a study showing the relative comparison of EEG and HRV under stressful condition (Surgery) to baseline EEG and HRV. Therefore, you can see the amount of change from baseline for each case. Therefore, it can be seen that the correlation between each other case is small compared to the baseline EEG and HRV performed before surgery. Due to the nature of the research conducted as a pilot study, there is a limit that cannot increase the number of surgeons. In this regard, it is described in limitation so that the reader can be aware of the clear limitations. Thanks for the sharp comment.
- The conclusions advanced in lines 323-328 seem to focus on differences between surgeons. This does not seem appropriate considering that only 8 surgeons (divided into 3 groups) were enrolled in the study.
- Thank you for your point. Only eight surgeons were enrolled in this study. However, I think that the number of surgical cases is not small with 265 cases. In this study, EEG and HRV parameters of each individual's baseline (resting status) were obtained for the first time. And, as the surgical stage progressed, both EEG and HRV parameters were measured in real time. And, it was averaged and divided by three stages to be quantified. Of course, the number of surgeons is small, and it cannot be said that there is no bias because the surgery performed by that surgeon. Therefore, in order to minimize this, we quantified the ratio between the parameters of resting status and the real time parameters from each surgical case. And, by dividing it into three groups divided according to surgical experience, this was an attempt to show how the ratio of stress-associated parameters according to time of each case in each group is different. Because it consists of a small number of surgeons like your comments, there are clearly limitations of the study. Therefore, this is specified in the limitation. Also, corrections have been made to clarify the meaning of conclusion. Thank you.
- The more experienced surgeons tend to be older. Aging is associated with baseline changes in HRV. Thus, there may be age-related changed that may affect baseline and hence responses during surgery. This may be worth addressing in the discussion.
- Thanks for your addressing. The content seems to have become more abundant.
- It seems unnecessary to include the last 3 rows of Table 1 considering that surgeons do not differ on these variables and is already mentioned in the text of the results. Rather, it may be useful to provide baseline values of heart rate and some of the stress measures.
- Thanks for your point. As per your point, I deleted the last 3 rows. And baseline values of HRV and EEG were recorded.
- Is the use of the words “real-time (assessment of stress)” accurate considering that a 5-min recording is used for HRV? Also, since the measure is not more instantaneous how do HRV values have utility for a surgeon?The data used in the analysis were measured throughout the operation time. That is, the analysis was performed using EEG and heart rate synchronized with the operation progress. Initially, a minimum of 256 heart beat peaks are required to analyze the heart rate. Considering that the average BPM of a normal person is 60-100, about 5 minutes of measurement data is required. 5-min recordings were not utilized as representative data of surgeon HRV.And we averaged all the EEG and HRV data corresponding to the 1st, 2nd, and 3rd stages. Therefore, we tried to reflect the surgeon's stress-associated parameter to the corresponding surgical stage as much as possible.Below is a reference for measuring HRV in the same way as above.
- Also, for analysis, data is not collected and analyzed every 5 minutes. 256 heart beats were needed for the initial analysis, and since then, whenever one peak appears, we tried to reflect the HRV in real time using the moving window method.
[1] Heart rate variability: standards of measurement, physiological interpretation and clinical use. Task Force of the European Society of Cardiology and the North American Society of Pacing and Electrophysiology. Circulation. 1996 Mar 1;93(5):1043-65. PMID: 8598068.
- What criteria was used in selecting a representative ECG segment? How were ectopic beats handled? Describing the criteria that were used is important to the validity of the research findings.As mentioned earlier, I did not select representative HRV parameters. All parameters of BPM, RMSSD, and LF/HR were measured in real time during the operation time through PPG (Photoplethysmography) sensor. And those data were averaged. The averaged values were rationed to the surgeon's baseline values. In doing so, I tried to minimize the possible biases during surgery with a small number of surgeons.
- The 2nd and 3rd sentence of the abstract should be edited for sentence structure.
- I have modified the 2nd and 3rd sentence of the abstract according to your comment.
- The first paragraph of the discussion does not seem to add much to this section and could be deleted. The 3rd paragraph of the discussion that begins with “The first main finding…” may be more effective if it moved up to the 1st paragraph of the discussion.
- I have made the flow of the discussion part smooth by deleting the first paragraph according to your opinion. Thanks for the point.
- In line 133 low RMSSD is said to reflect increased stress whereas in line 139 stress is accompanied by increased RMSSD.
- According to your opinion, there was an incorrectly described part and a correction was made. Thank you for your point.
- A reference that supports the validity of using earlobe pulse for heart rate variability in short-term recordings would be important to include in the methods. Are their limitations to address with this technique such as influence of respiratory rhythm?Based on your comment, I have attached a reference that discusses the usefulness of HRV detecting device using earlobe pulse. This paper measured the surgeon's HRV over a long period (average operation time: over 120mins). Therefore, as the reviewer noted, respiratory rhythm can influence the data. Also, since it is a measure of long-term heart rate variability, there may be an effect, but the effect seems to be insignificant compared to other detecting devices. Therefore, the following references have been added to the reference list.[14] Vescio B, Salsone M, Gambardella A, Quattrone A. Comparison between Electrocardiographic and Earlobe Pulse Photoplethysmographic Detection for Evaluating Heart Rate Variability in Healthy Subjects in Short- and Long-Term Recordings. Sensors (Basel). 2018;18(3):844. Published 2018 Mar 13. doi:10.3390/s18030844
- References ‘1’ and ‘2’ needs revision in the references list.
- I have modified references list according to your comment.
- Were there unanticipated events that occurred during any of the surgeries? What was the associated change in HRV and EEG patterns under these conditions? This may provide additional strength to the role of surgeon experience and ‘staying cool under pressure’.
- Thanks for your comment. Unanticipated events (iatrogenic nerve/blood vessel damage) and changes in HRV and EEG patterns were considered in the initial design of the study. However, in this study, due to heterogeneity of surgical procedures with other subspecialty, consistent data were not available. Therefore, after dividing the operation according to the tourniquet use, it was thought that operative time & intraoperative blood loss, which can be regarded as the major among the surgical factors predicted by the surgeon, can be assumed as a widespread unanticipated event. As a result, it was found that the stress-associative parameters were significantly correlated with blood loss and operative time. According to your comment, future research will measure the changes in EEG and HRV patterns according to specific events. Thanks for the comment.

Round 2
Reviewer 1 Report
The authors have made mostly the suggested changes, and show better the differences with the article with which I had concerns.
However, there seems to have been confusion with my earlier comment about limitations on the number of participants and their segmentation. I was referring to the authors stating in the paper that the small number of participants is a limitation in itself, and that an attempt will be made to increase the size of the group in the future.
The only remarkable thing this time, is that I would suggest to change the text in the image of figure 1 from "EEG-left" and "EEG-right" to FP1 and FP2.
Author Response
The authors have made mostly the suggested changes, and show better the differences with the article with which I had concerns.
However, there seems to have been confusion with my earlier comment about limitations on the number of participants and their segmentation. I was referring to the authors stating in the paper that the small number of participants is a limitation in itself, and that an attempt will be made to increase the size of the group in the future.
The only remarkable thing this time, is that I would suggest to change the text in the image of figure 1 from "EEG-left" and "EEG-right" to FP1 and FP2.
I would like to express my appreciation for your efforts in reviewing our manuscript.
I changed the text in the image of figure 1 according to your comment.

Reviewer 2 Report
The revisions to the manuscript are appreciated. The authors have sufficiently addressed my comments.
Author Response
The revisions to the manuscript are appreciated. The authors have sufficiently addressed my comments.
Thank you for your enormous efforts in the reviewing process.

This manuscript is a resubmission of an earlier submission. The following is a list of the peer review reports and author responses from that submission.